# Comprehensive Clinical and Genetic Analysis of *CHEK2* in Croatian Men with Prostate Cancer

**DOI:** 10.3390/genes13111955

**Published:** 2022-10-27

**Authors:** Kira Kirchner, Marija Gamulin, Tomislav Kulis, Bianca Sievers, Zeljko Kastelan, Davor Lessel

**Affiliations:** 1Institute of Human Genetics, University Medical Center Hamburg-Eppendorf, 20246 Hamburg, Germany; 2Department of Oncology, University Hospital Center Zagreb, University of Zagreb School of Medicine, 10000 Zagreb, Croatia; 3Department of Urology, University Hospital Centre Zagreb, University of Zagreb School of Medicine, 10000 Zagreb, Croatia

**Keywords:** CHEK2, prostate cancer, aggressive prostate cancer

## Abstract

Germline pathogenic and likely pathogenic (P/LP) variants in *CHEK2* have been associated with increased prostate cancer (PrCa) risk. Our objective was to analyze their occurrence in Croatian PrCa men and to evaluate the clinical characteristics of P/LP variant carriers. Therefore, we analyzed *CHEK2* in 150 PrCa patients unselected for age of onset, family history of PrCa or clinical outcome, and the frequency of identified variants was compared to findings in 442 cancer-free men, of Croatian ancestry. We identified four PrCa cases harboring a P/LP variant in *CHEK2* (4/150, 2.67%), which reached a statistical significance (*p* = 0.004) as compared to the control group. Patients with P/LP variants in *CHEK2* developed PrCa almost 9 years earlier than individuals with *CHEK2* wild-type alleles (8.9 years; *p* = 0.0198) and had an increased risk for lymph node involvement (*p* = 0.0047). No association was found between *CHEK2* status and further clinical characteristics, including the Gleason score, occurrence of aggressive PrCa, the tumor or metastasis stage. However, carriers of the most common P/LP CHEK2 variant, the c.1100delC, p.Thr367Met*fs*15*, had a significantly higher Gleason score (*p* = 0.034), risk for lymph node involvement (*p* = 0.0001), and risk for developing aggressive PrCa (*p* = 0.027). Thus, in a Croatian population, *CHEK2* P/LP variant carriers were associated with increased risk for early onset prostate cancer, and carriers of the c.1100delC, p.Thr367Met*fs*15* had increased risk for aggressive PrCa.

## 1. Introduction

*CHEK2* (*Checkpoint kinase 2*), encoding the CHK2 protein, is a serine/threonine protein kinase and one of the key components of the DNA damage response (DDR) pathway. CHK2 has pleiotropic functions regulating cell cycle progression after endogenous or exogenous DNA damage or the blockade of DNA replication, thereby preventing the entry of damaged cells into mitosis [1]. Given its biological role, it is not unexpected that genetic studies have implicated *CHEK2* as a multiorgan tumor susceptibility gene. Indeed, *CHEK2* has been associated with an increased risk for testicular germ-cell tumors, breast cancer, and colorectal cancers [2,3,4].

Moreover, *CHEK2* has been implicated in conferring an increased risk of prostate cancer (PrCa) [5], especially early onset [6] and metastatic PrCa [7]. In light of the therapeutic actionability of pathogenic and likely pathogenic (P/LP) *CHEK2* variants in metastatic castration-resistant prostate cancer (mCRPC) [8], the identification of germline variants that not only predict the disease risk, but also have implications for the clinical outcome, has increasing relevance, especially since recent large-scale analyses have demonstrated that P/LP variants in cancer susceptibility genes are identified in more than 50% of the cases who do not meet formal germline testing criteria [9].

Prompted by our recent findings where we established *CHEK2* as the first moderate-penetrance testicular germ-cell tumor (TGCT) susceptibility gene and identified the low-penetrance *CHEK2* p.Ile157Thr variant as a likely founder missense variant in Croatian men [2], this study aimed to evaluate the PrCa risk conferred by the P/LPs in *CHEK2* along with its clinical implications in a Croatian population.

## 2. Materials and Methods

### 2.1. Study Participants

A total of 150 Croatian patients with a primary prostate cancer (Appendix A), unselected for age of onset, family history of PrCa or clinical outcome, were ascertained within a single year by the Departments of Oncology and Urology at the Clinical Hospital Centre Zagreb, Croatia. Clinical data on all cases were uniformly obtained from an oncology specialist using a structured clinical table. Aggressive cases (n = 25) were defined as previously described [10]. Briefly, the following criteria defined an aggressive PrCa: (i) cause of death (regardless of T-stage or Gleason score), (ii) stage 4 (regardless of T-stage or Gleason score), or (iii) T-stage 3 and Gleason score ≥ 8. The control group consisted of 442 cancer-free Croatian men in whom we had previously analyzed the occurrence of P/LP variants in *CHEK2* [2]. Written informed consent was obtained from each individual after a detailed explanation of the purpose of the study. The study was approved by the ethics committee of the Clinical Hospital Centre Zagreb, Zagreb, Croatia, 8.1-19/125-3.

### 2.2. Genetic Analyses

The extraction of genomic DNA from whole-blood samples was performed by a standard salting-out method, as described previously [11]. A PCR amplification and Sanger sequencing of all 14 coding exons of *CHEK2* was performed as previously described [2]. In addition to the previously reported protocols for the detection of exon 9_10 deletion [2], the occurrence of this deletion was verified by PCR amplification as described previously [12]. The clinical interpretation of identified *CHEK2* variants followed the American College of Medical Genetics and Genomics (ACMG) criteria, as previously described [2].

### 2.3. Statistical Analyses

For the statistical analysis, an unpaired *t*-test or a Fischer’s exact test were performed, using GraphPad Prism 8.

## 3. Results

A detailed clinicopathological characterization of the 150 Croatian prostate cancer (PrCa) patients included in the study is shown in Table 1 and in more detail in Appendix A. In summary, the mean age at onset of PrCa was 68.17 years (ranging from 48 to 84 years). Eighteen cases developed additionally another tumor (12%). Regarding the PrCa, the mean size of the primary tumor (T-stage) was 2.10, 18 cases developed a primary metastasis (12%), and in 4 cases the PrCa was spread to lymph nodes (2.7%), whereas the mean Gleason score was 6.99 (ranging from 4 to 10). Twenty-five cases developed aggressive PrCa (16.7%), out of which eleven deceased due to PrCa-related complications (7.3%). A family history of tumor occurrence was available for only 119 PrCa patients (79.3%). Notably, none had a family history of PrCa and six had a family history of any malignancy (5%). Out of the latter, three had a mother who had developed a breast cancer (2.5%).

Utilizing a direct Sanger sequencing approach followed by a PCR amplification for the detection of exon 9_10 deletion, we identified pathogenic or likely pathogenic (P/LP) *CHEK2* germline variants in four PrCa cases (4/150, 2.67%). In more detail, two cases harbored the c.1100delC, p.Thr367Metfs15*, and two cases harbored the c.1169A>C, p.Tyr390Ser (Figure 1). Comparing these data to the control group [2], the difference reached statistical significance (4/150 vs. 0/442, *p* = 0.004; Table 2). We additionally identified two cases harboring the low-penetrance variant c.470T>C, p.Ile157Thr. However, contrary to our findings in TGCT’s [2], the p.Ile157Thr among Croatian men with PrCa was statistically insignificant compared with ethnically matched noncancer controls (2/150 vs. 6/442, *p* = 0.99; Table 2). Moreover, we identified a single PrCa case harboring c.1270T>C, p.Y424H, a variant classified as a variant of unknown significance (Figure 1).

We next performed a case–case analysis comparing the clinical characteristics of PrCa cases in our study, stratified by *CHEK2* variant status. The mean age of onset for P/LPs carriers was 59.5 ± 6.75 compared to 68.4 ± 0.60 in noncarriers, a finding that reached statistical significance (8.9 years difference; *p* = 0.0198). Moreover, P/LP carriers had a statistically increased risk for lymph node involvement (0.25 ± 0.25 vs. 0.02 ± 0.01, *p* = 0.0047). A further comparison regarding the Gleason score, the occurrence of aggressive PrCa, at the tumor and metastasis stage, all revealed statistically insignificant *p*-values (Table 3). We further compared the clinical characteristics of the two carriers of the most common P/LP *CHEK2* variant, c.1100delC, p.Thr367Metfs15* [13], to our PrCa cohort. The carriers of this variant had a 10-year earlier age of onset (58.0 ± 10 vs. 68.3 ± 0.61), a finding which reached a borderline significance (*p* = 0.056). Moreover, there was a statistically significant enrichment regarding the development of aggressive PrCa by the carriers of this variant (2/25, 8% vs. 0/125, 0%, *p* = 0.027), and the carriers had a higher Gleason score (8.5 ± 0.5 vs. 6.97 ± 0.08, *p* = 0.034) and increased risk for lymph node involvement (0.5 ± 0.5 vs. 0.02 ± 0.01, *p* < 0.0001).

## 4. Discussion

Prostate cancer (PrCa) is one of the most common male cancers worldwide, the second tumor-associated cause of death in men, and one of the most heritable human cancers [14,15]. In line with the strong heritability of PrCa, genome-wide association studies (GWAS) have reported more than 260 common low-penetrance genetic variants associated with PrCa risk [16]. Moreover, pathogenic or likely pathogenic germline variants (P/LPs) in several susceptibility genes have been shown to confer a moderate-to-high increase in risk for PrCa [17]. In the past, the analysis of the germline susceptibility genes was primarily utilized to identify individuals at high risk for developing cancer, who would benefit from enhanced cancer surveillance and early screening strategies. However, recent studies established the additional utility of germline genetic testing in PrCa both for the prediction of the clinical outcome [7,10] and for germline-directed treatment [8].

P/LPs variants in the tumor suppressor gene *CHEK2* have been associated with increased PrCa risk [5,18,19] and belong to the group of germline variants with therapeutic actionability [8]. Previous studies, analyzing the implications of *CHEK2* P/LP variants for clinical care have utilized cohorts enriched for cases with metastatic PrCa [7], lethal PrCa [13,20], aggressive PrCa [10], early onset [6], or primarily analyzing ethnic-specific founder effects [21].

Here, we performed the first comprehensive *CHEK2* analysis in Croatian men with PrCa that were unselected for the age of onset, family history of PrCa, or its clinical outcome. According to recent epidemiological data, we included around 10% of all patients in Croatia who developed PrCa within one year [22,23]. We found *CHEK2* P/LP in 2.67% of unselected PrCa cases that were significantly enriched compared to the control group. Thus, in line with previous studies [5,6,7,12], our findings further support *CHEK2* being a moderate PrCa susceptibility gene. Moreover, despite the rather small sample size, our findings may be informative for the clinical management of PrCa cases, at least in Croatia. Namely, the cases harboring a *CHEK2* P/LP presented almost 9 years earlier than noncarriers and had an increased risk for lymph node involvement. Furthermore, we identified two recurrent P/LPs, c.1100delC, p.Thr367Met*fs*15*, and c.1169A>C, p.Tyr390Ser, that should be included in the local PrCa germline screening, as both of these variants enable a genotype-directed treatment. Interestingly, out of these two P/LPs only the c.1100delC, p.Thr367Met*fs*15* was associated with aggressive PrCa, a finding previously observed in PrCa cases of European ancestry [13]. Moreover, we found that carriers of this variant had a significantly higher Gleason score, and similarly to overall P/LP carriers, a statistically increased risk for lymph node involvement.

One of the limitations of this study is the absence of familial PrCa cases along with cases having a family history of any malignancy. It is, however, worth noting that the mother of one of the cases harboring a P/LP variant developed breast cancer. Although her DNA was not available for testing, a recent study suggested that a family history of breast cancer and high Gleason score could be predictors to identify PrCa cases harboring P/LP germline variants in one of the DNA repair genes [24].

In addition, the low-penetrance variant c.470T>C, p.Ile157Thr, was not overrepresented in Croatian PrCa cases. Thus, our data might point to p.Ile157Thr being a specific risk factor for the testicular germ-cell tumors (TGCT) in the Croatian population.

## 5. Conclusions

Taken together, our study suggested *CHEK2* P/LPs as a moderate PrCa-susceptibility gene associated with early age of onset in the Croatian population. In addition, we provided further evidence for the association of the c.1100delC, p.Thr367Met*fs*15* with aggressive PrCa.

## Figures and Tables

**Figure 1 genes-13-01955-f001:**
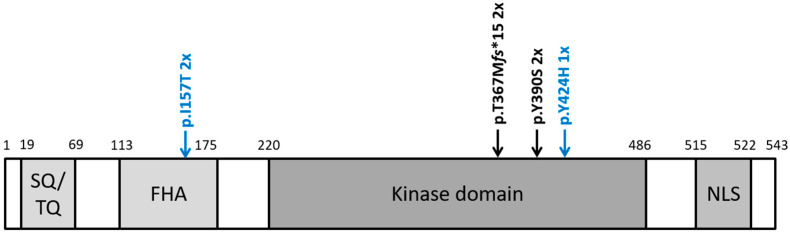
Location of identified *CHEK2* variants. Schematic protein structure of CHK2 showing conserved domains. SQ/TQ: SQ/TQ cluster domain; FHA: forkhead-associated domain; kinase domain; NLS: nuclear localization signal. Pathogenic and likely pathogenic variants are shown in black above the protein structure; the other variants are shown in blue.

**Table 1 genes-13-01955-t001:** Clinicopathological characteristics of the 150 patients included in this study.

Status Total	Cases n = 150
Mean age at diagnosis (mean +/− SEM; range)	68.17 (+/−0.62; 48–84)
Cases with other tumors (%)	18 (12)
T-stage (%)	
T1	26 (17.33)
T2	83 (55.33)
T2/T3	2 (1.33)
T3	32 (21.33)
T4	4 (2.66)
Unknown	3 (2)
Mean (+/−SEM; range)	2.10 (+/−0.06)
Presence of metastasis (M-stage; %)	Yes: 132 (88)
	No: 18 (12)
Spread to lymph nodes (N-stage; %)	Yes: 142 (94.67)
	No: 4 (2.67)
	Unknown: 4 (2.67)
Gleason score (%)	
4	1 (0.67)
6	51 (34)
7	58 (38.67)
8	23 (15.33)
9	10 (6.67)
10	3 (2)
Unknown	4 (2.67)
Mean (+/−SEM; range)	6.99 (+/−0.08; 4–10)
Aggressive PrCa cases (%)	25 (16.7)
Death due to PrCa (%)	11 (7.3)
Family history of prostate cancer (available for cases)	0 (119)
Family history of any tumor (available for cases; %)	6 (119; 5)

**Table 2 genes-13-01955-t002:** Enrichment analysis: frequency of *CHEK2* variants identified in Croatian prostate cancer patients compared to cancer-free Croatian men. P/LPs: pathogenic and likely pathogenic variants (c.1100delC, p.T367Mfs*15; c.1169A>C, p.Y390S). Significant *p*-values are depicted in bold.

*CHEK2* Variants	Genotype Count		OR (95% CI) Per Allele	*p*-Value
	Controls	Cases		
P/LPs	0/0/442	0/4/146	infinity (2.97 to infinity)	**0.004**
c.470T>C, p.I157T	0/6/436	0/2/148	0.98 (0.2–4.08)	0.99

**Table 3 genes-13-01955-t003:** Clinicopathological characteristics of carriers of P/LP CHEK2 variants and noncarriers. Significant *p*-values are depicted in bold.

	*CHEK2* WT	*CHEK2* P/LP Carriers	*p*-Value	*CHEK2* non c.1100delC Carriers	*CHEK2* c.1100delC Carriers	*p*-Value
Age of diagnosis	68.4 ± 0.60	59.5 ± 6.75	**0.0198**	68.3 ± 0.61	58.0 ± 10	0.0556
Gleason score	6.97 ± 0.09 (142/144 patients)	7.5 ± 0.65	0.3070	6.965 ± 0.08	8.5 ± 0.5	**0.0336**
T-stage	2.10 ± 0.06 (143/144 patients)	2.25 ± 0.48	0.6728	2.09 ± 0.06	3	0.0704
M-stage	0.12 ± 0.03	0	0.4575	0.12 ± 0.03	0	0.6020
N-stage	0.02 ± 0.01	0.25 ± 0.25	**0.0047**	0.02 ± 0.01	0.5 ± 0.5	**<0.0001**
Aggressive PrCa	23/146	2/4	0.2960	23/148	2/2	**0.0268**

## Data Availability

Not applicable.

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
