# Peer review of "Comprehensive Clinical and Genetic Analysis of CHEK2 in Croatian Men with Prostate Cancer"

_genes, 2022, doi:10.3390/genes13111955_

Round 1

Reviewer 1 Report

1. Another publication pointing to CHEK2 P / LPs variants as a moderate PrCa susceptibility. An important and also earlier reported observation, that prostate cancers caused by CHEK2 mutations appear earlier. The Croatian study confirms this observation.

2. The discussion could include a wider range of works on this subject, especially related to the European population, e.g.:

Cybulski C, Huzarski T, Górski B, Masojć B, Mierzejewski M, Debniak T, Gliniewicz B, Matyjasik J, ZÅ‚owocka E, Kurzawski G, Sikorski A, Posmyk M, Szwiec M, Czajka R, Narod SA, LubiÅ„ski J. A novel founder CHEK2 mutation is associated with increased prostate cancer risk.Cancer Res. 2004 Apr 15;64(8):2677-9.

Seppala EH, Ikonen T, Mononen N, et al CHEK2 variants associate with hereditary prostate cancer. Br J Cancer89: 1966-70,  2003.

However, this is not a complaint but a suggestion.

3. A very interesting issue is the answer to the question whether there was a loss of the wild-type CHEK2 allele (LOH) in any of prostate cancers from men who carried CHEK2 mutations? The question is whether the researchers will have the determination to get to the tumor material ...

Author Response

  1. Another publication pointing to CHEK2 P / LPs variants as a moderate PrCa susceptibility. An important and also earlier reported observation, that prostate cancers caused by CHEK2 mutations appear earlier. The Croatian study confirms this observation.

Response 1: We thank the reviewer for their interest and their thoughtful comments and feedback.

  1. The discussion could include a wider range of works on this subject, especially related to the European population, e.g.:

Cybulski C, Huzarski T, Górski B, Masojć B, Mierzejewski M, Debniak T, Gliniewicz B, Matyjasik J, ZÅ‚owocka E, Kurzawski G, Sikorski A, Posmyk M, Szwiec M, Czajka R, Narod SA, LubiÅ„ski J. A novel founder CHEK2 mutation is associated with increased prostate cancer risk.Cancer Res. 2004 Apr 15;64(8):2677-9.

Seppala EH, Ikonen T, Mononen N, et al CHEK2 variants associate with hereditary prostate cancer. Br J Cancer, 89: 1966-70,  2003.

However, this is not a complaint but a suggestion.

Response 2: As suggested, the abovementioned studies have been cited in the revised version of the manuscript.

  1. A very interesting issue is the answer to the question whether there was a loss of the wild-type CHEK2 allele (LOH) in any of prostate cancers from men who carried CHEK2 mutations? The question is whether the researchers will have the determination to get to the tumor material

Response 3: We fully agree with the reviewer that this is a very interesting question. Indeed, the literature makes note to somewhat contradictory findings regarding LOH in tumors with a germline CHEK2 mutation. However, the tumors samples of the mutation carriers identified in this study are unfortunately not available, thus we were not able to address this interesting question.

Reviewer 2 Report

In the article entitled “Comprehensive clinical and genetic analysis of CHEK2 in Croatian men with prostate cancer”, the authors aimed to study the occurrence of germline pathogenic and likely pathogenic (P/LP) variants in CHEK2 gene in Croatian patients with increased prostate cancer (PrCa) risk and to evaluate the clinical characteristics of P/LP variant carriers. They identified statistical significant PrCa cases harboring a P/LP variant in CHEK2 and concluded that in Croatian population, CHEK2 P/LP variant carriers are associated with increased risk for early-onset as well as aggressive prostate cancer.

These data may lead to new potential basis for the prognosis and treatment of this disease.

The paper is sufficiently clear to read but it needs to be slightly revised before it could be considered for publicaton, as reported here:

The authors discussed about the role of germline mutations in CHEK2 gene. They reported that “CHEK2, encoding the CHK2 protein, is a serine/threonine protein kinase and one of the key components of the DNA damage response (DDR) pathway”, but they didnt’ reported what CHEK2 stay for. Please indicate the name of the gene.

In the MATHERIALS and METHODS Section, sub-paragraph named “Study participants”, the authors reported that “The study was approved by the ethics committee of the Clinical Hospital Centre Zagreb, Zagreb, Croatia”, but they didn’t reported protocol authorization number. Please absolutely add it to the text.

This is a Brief Report which elucidate the possible correlation between germline pathogenic and likely pathogenic (P/LP) variants in CHEK2 gene with increased prostate cancer (PrCa) risk, in Croatian patients. It would be interesting to verify by functional assays the molecular consequences of these pathogenic variants on prostate cells behaviour, in a more complete and scientifically interesting manuscript.

In this paper, the authors only reported some interesting data about the involvement of pathogenic and likely pathogenic (P/LP) variants in CHEK2 gene in prostate cancer onset, suggesting a potential prognostic and therapeutic application for this. By now, I suggest the authors to lead functional assays to support their conclusions reached in this brief report.

The data are interesting and clearly explained in the text. The paper potentially reports interesting scientific data and meet the journal aims and to be considered for publication., as a Brief Report. 

Author Response

Reviewer 2:

In the article entitled “Comprehensive clinical and genetic analysis of CHEK2 in Croatian men with prostate cancer”, the authors aimed to study the occurrence of germline pathogenic and likely pathogenic (P/LP) variants in CHEK2 gene in Croatian patients with increased prostate cancer (PrCa) risk and to evaluate the clinical characteristics of P/LP variant carriers. They identified statistical significant PrCa cases harboring a P/LP variant in CHEK2 and concluded that in Croatian population, CHEK2 P/LP variant carriers are associated with increased risk for early-onset as well as aggressive prostate cancer. These data may lead to new potential basis for the prognosis and treatment of this disease.

We thank the reviewer for their interest and their thoughtful comments and feedback. Below are our responses to the questions and concerns raised by the reviewer.

The paper is sufficiently clear to read but it needs to be slightly revised before it could be considered for publicaton, as reported here:

  1. The authors discussed about the role of germline mutations in CHEK2 gene. They reported that “CHEK2, encoding the CHK2 protein, is a serine/threonine protein kinase and one of the key components of the DNA damage response (DDR) pathway”, but they didnt’ reported what CHEK2 stay for. Please indicate the name of the gene.

 Response 1: As suggested, we have now included in the introduction that CHEK2 stand for “Checkpoint kinase 2”.

  1. In the MATHERIALS and METHODS Section, sub-paragraph named “Study participants”, the authors reported that “The study was approved by the ethics committee of the Clinical Hospital Centre Zagreb, Zagreb, Croatia”, but they didn’t reported protocol authorization number. Please absolutely add it to the text.

Response 2: We thank this reviewer for careful reading; Indeed, we have missed to include the protocol number (8.1-19/125-3), which has now been added.

  1. This is a Brief Report which elucidate the possible correlation between germline pathogenic and likely pathogenic (P/LP) variants in CHEK2 gene with increased prostate cancer (PrCa) risk, in Croatian patients. It would be interesting to verify by functional assays the molecular consequences of these pathogenic variants on prostate cells behaviour, in a more complete and scientifically interesting manuscript.In this paper, the authors only reported some interesting data about the involvement of pathogenic and likely pathogenic (P/LP) variants in CHEK2 gene in prostate cancer onset, suggesting a potential prognostic and therapeutic application for this. By now, I suggest the authors to lead functional assays to support their conclusions reached in this brief report.

Response 3: Although both missense variants reported in this study, p.Ile157Thr and p.Tyr390Ser, have been extensively studied before, we fully agree with this reviewer that it would be additionally interesting to study their behavior in a prostate and/or prostate cancer cell line. However, we currently do not have any such cell line in our lab. Moreover, such studies will require extensive further work and will thus remain an important aspect of our future studies.

  1. The data are interesting and clearly explained in the text. The paper potentially reports interesting scientific data and meet the journal aims and to be considered for publication., as a Brief Report.

Response 4: We thank the reviewer again for their interest and their thoughtful comments and feedback.